# Biotechnological Approach for the Production of Enantiomeric Hydroxylactones Derived from Benzaldehyde and Evaluation of Their Cytotoxic Activity

**Marcelina Mazur [1],***, **Anna Kudrynska [1]**, **Aleksandra Pawlak [2]**, **Beatriz Hernandez-Suarez [2]**, **Bożena Obmińska-Mrukowicz [2]** and **Witold Gładkowski [1]**

[1] Department of Chemistry, Wrocław University of Environmental and Life Sciences, Norwida 25, 50-375 Wrocław, Poland; aaahhaaa@ukr.net (A.K.); witold.gladkowski@upwr.edu.pl (W.G.)

[2] Department of Pharmacology and Toxicology, Wrocław University of Environmental and Life Sciences, Norwida 31, 50-375 Wrocław, Poland; aleksandra.pawlak@upwr.edu.pl (A.P.); beatriz.hernandez-suarez@upwr.edu.pl (B.H.-S.); bozena.obminska-mrukowicz@upwr.edu.pl (B.O.-M.)

* Correspondence: marcelina.mazur@upwr.edu.pl; Tel.: +48-713-205-197

**Abstract:** The β-aryl-δ-halo-γ-lactones are known for their antiproliferative activity towards numerous cancer cell lines. The aim of this study was to obtain in the biotransformation process new β-aryl-δ-hydroxy-γ-lactones and compare their activity with the antiproliferative activity of parent compounds. The racemic *cis*-5-(1-iodoethyl)-4-phenyldihydrofuran-2-one as well as separate enantiomers were transformed in fungal cultures. Among ten tested biocatalysts, three (*Absidia cylindrospora* AM336, *Absidia glauca* AM254, and *Fusarium culmorum* AM10) were able to catalyze the hydrolytic dehalogenation process. The biotransformations processes were highly stereoselective and enantiomerically pure hydroxylactones were obtained (ee ≥ 99%). The iodo- and hydroxylactone enantiomers were subjected to cytotoxic activity evaluation on canine leukemia and lymphoma cell lines. The iodolactones exhibited higher biological potential towards tested cell lines than hydroxylactones. Higher cytotoxic potential was also characteristic for (+)-(4*S*,5*S*,6*R*)-enantiomer of iodolactone compared to its antipode.

**Keywords:** lactones; biotransformations; dehalogenation; antiproliferative activity

## 1. Introduction

Lactones are a group of compounds widely present in nature. Those cyclic esters are generally produced as a plant metabolite and often exhibit interesting biological properties. Lactones can be sourced from numerous plants used in folk medicine. Vithanolide A, from *Withania somnifera*, the traditional Ayurvedic plant, is known for neurone regeneration potential [1]. Anti-malarial artemisinin is isolated from *Artemisia annua*, used in folk Chinese medicine [2,3]. Anti-viral brevilin A is sourced from medicinal herb *Centipeda minima* [4]. Among the naturally occurring lactones, the special interests are put on those with cytotoxic and anticancer activity. The lactones containing aromatic ring-like styryl lactones, lignan lactones and β-aryl-γ-lactones are often discussed in context of drug development [5–7]. Unfortunately, one of most important aspects limiting the wider use of naturally occurring anticancer compounds is their low concentration in natural sources. Therefore, in numerous works, the total synthesis of those compounds is presented as well as their chemical modifications [8,9]. This approach involves certain disadvantages such as using the harsh chemicals, difficulties with obtaining enantiomerically pure product and hence the use of complex and expensive

catalysts. Biotransformation processes can be interesting alternatives to traditional chemical synthesis in the context of sourcing new biologically active compounds. Microbial transformations are considered to be highly regio- and stereoselective, require mild reaction conditions and are more environmentally friendly. Biotransformations are also a great tool for metabolism investigations. The fungal transformations often serve as a model processes for mimicking the metabolic pathways occurring in higher organisms [10,11]. The applications of biotransformation can be crucial for the determination of metabolite structures, which are formed in small concentrations and are often further transformed quickly into low-molecular-weight products. The use of appropriate fungal catalysts can lead to the accumulation of metabolites and facilitate their isolation and identification. The biotransformation processes are also important for sourcing new biologically active compounds [12–14].

In our previous study, we developed the synthetic approach to obtain β-aryl-γ-lactones containing a halogen atom that exhibit significant antiproliferative activity towards tested cancer cell lines [15–18]. In this work we adapt the biotransformation process to the preliminary investigation of their metabolism in fungal cultures as well as to obtain the new hydroxylactones as potential cytotoxic agents.

## 2. Results and Discussion

The chiral iodolactones **1a** and **1b** were obtained by chemoenzymatic synthesis developed to access the β-aryl-δ-halo-γ-lactones. The absolute configurations of the stereogenic centers of the synthesized enantiomers were determined earlier on the basis of crystallographic analysis [15]. The racemic form of iodolactone (*rac*-**1**) exhibited cytotoxicity towards acute human leukemia (Jurkat) and canine osteosarcoma (D17) cell lines [16]. Based on those findings, we planned to further investigate the metabolism of iodolactone enantiomers **1a** and **1b** as well as determine the biological potential of the obtained products and the chiral form of substrate.

The metabolism of halogenated organic compounds may occur in several pathways and most of them include different types of dehalogenation reactions. The dehalogenation processes are often one of the first steps which allow for further transformation and detoxification of xenobiotics. Depending on the environmental conditions and type of microorganism used, different dehalogenation processes can be involved, for example, reductive dehalogenation, dehydrohalogenation and hydrolytic dehalogenation [19–21]. In our research, we chose different strains of fungi and the screening procedure allowed to select the microorganism able to transform iodolactone (*rac*-**1**). In the literature, examples of different halolactones biotransformations are presented. Those processes often lead to hydrolytic dehalogenation products [22–24]. In the presented study, regardless of the applied biocatalyst, only one product with different efficiency was formed. The effective biocatalysts were *Absidia cylindrospora* AM336, *Absidia glauca* AM254 and *Fusarium culmorum* AM10. Other tested strains did not exhibit the ability for substrate transformation. In order to isolate the product and determine its structure, the biotransformation was performed on a multiplied scale. Based on the data presented in Figure 1, the process was performed using *Absidia glauca* AM254, due to the highest substrate conversion.

The structure of isolated product was determined on the basis of spectroscopic data. The IR spectrum of the product shows a strong absorption band for the hydroxyl group (3442 cm$^{-1}$) and also for a carbonyl group in the γ-lactone ring (1772 cm$^{-1}$), which indicates the formation of hydroxylactone **2** (Figure S1). The NMR data also confirm those findings. The $^{13}$C NMR spectrum shows the difference in the chemical shift of signals from the C-6 carbon atom (Figure S2). The position of the C-6 signal at δ = 66.8 ppm confirms the presence of a hydroxyl group in the product molecule. In the spectrum of iodolactone, the C-6 carbon atom is present at 23.3 ppm, which is caused by the occurrence of the heavy atom effect characteristic for carbon bounded with an iodine atom [25].

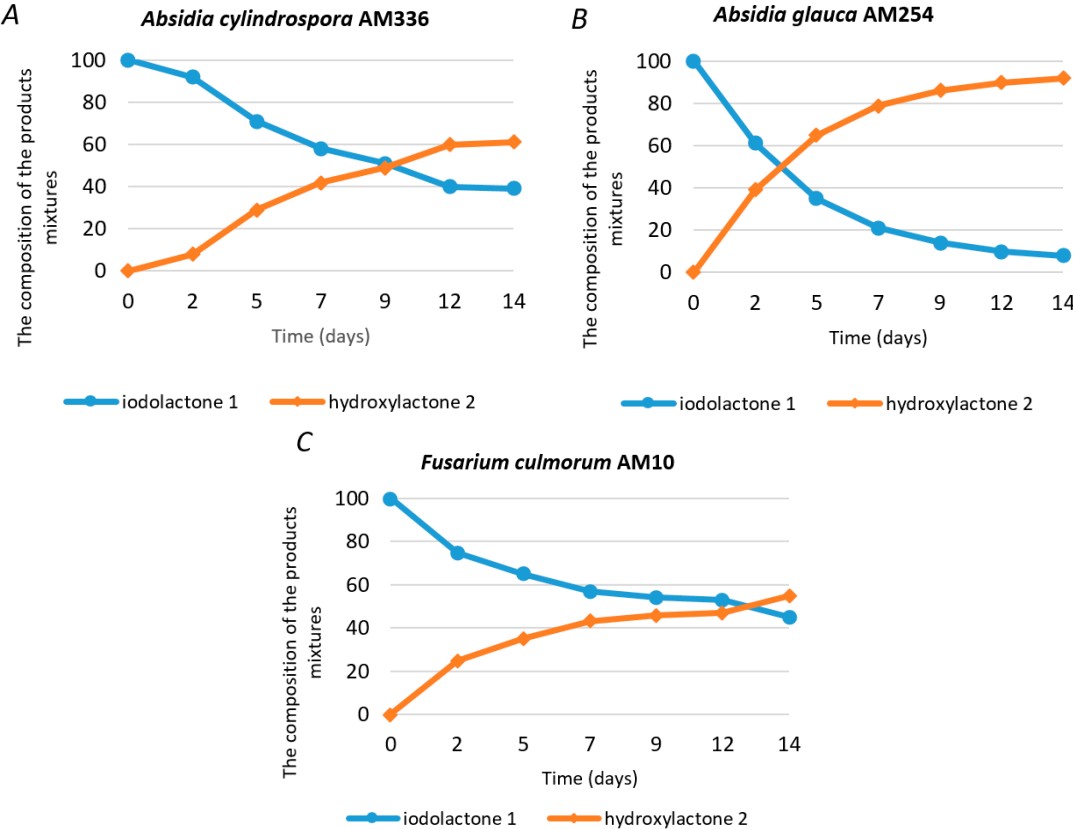

**Figure 1.** Composition (in % according to GC) of the product mixtures in the screening biotransformations of racemic iodolactone (*rac*-**1**) by different fungal strains. (**A**) *Absidia cylindrospora* AM336; (**B**) *Absidia glauca* AM254 and (**C**) *Fusarium culmorum* AM10.

The [1]H NMR spectrum of hydroxylactone **2** (Figure S3) is clearly distinct from the iodolactone **1** spectrum. The first difference relates to the chemical shift of methyl protons: δ = 2.01 ppm for iodolactone and δ = 1.13 ppm for hydroxylactone. The similar tendency can be observed for the H-5 proton which also is in α-position towards the hydroxyl group. On the product **2** spectrum, the H-5 proton is shifted slightly upfield to 4.49 ppm compared to iodolactone (4.82 ppm). The most important difference can be noticed in the H-6 proton coupling constants. This change directly indicates the shift in angles between the vicinal protons H5 and H6 and suggests the inversion of the configuration on the C-6 carbon atom. On the iodolactone spectrum, the H-6 signal is present as a doublet of quartets at 3.47 ppm. On the spectrum of product **2**, this signal appears as a quartet of doublets at 3.58 ppm because of the significantly lower coupling constant between H-6 and H-5 ($J$ = 4.3 Hz). The high value of the coupling constant in iodolactone **1** ($J$ = 10.8 Hz) indicates the antiperiplanar orientation of protons H-5 and H-6 [15]. Contrastingly, the significantly lower value of this coupling constant in the hydroxylactone spectrum indicates the reduction of the dihedral angle, which is related to the formation of diastereoisomer in $S_N2$ type of substitution. The structure of the obtained product was also confirmed by the heteronuclear multiple-bond correlation (HMBC) correlation spectrum (Table 1, Figure S4).

Interestingly, the NMR data are also significantly different than those reported for the *trans*- diastereoisomer of hydroxylactone obtained earlier by the chemical lactonization of unsaturated ester with *m*-chloroperbenzoic acid [26]. The *trans*-diastereoisomer was isolated from a mixture of products, with rather low (9%) yield for chemical synthesis, whereas *cis*-5-(1-hydroxyethyl)-4-phenyldihydrofuran-2-one **2** obtained in the biotransformation process was isolated in 38% yield.

**Table 1.** Heteronuclear multiple-bond correlation (HMBC) data of hydroxylactone **2**.

|  | CH$_2$-3 | H-4 | H-5 | H-6 | CH$_3$-7 | H-2′; H-6′ | H-3′; H-5′ | H-4′ |
|---|---|---|---|---|---|---|---|---|
| C-2 | + | + | + |  |  |  |  |  |
| C-3 |  | + | + |  |  |  |  |  |
| C-4 | + |  | + |  |  | + |  |  |
| C-5 | + | + |  |  | + |  |  |  |
| C-6 |  |  |  |  |  |  |  |  |
| C-7 |  |  | + | + |  |  |  |  |
| C-1′ | + | + | + |  |  |  | + |  |
| C-2′; C-6′ |  | + |  |  |  | + |  | + |
| C-3′; C-5′ |  |  |  |  |  |  | + |  |
| C-4′ |  |  |  |  |  | + |  |  |

The NMR data were also related to the product obtained in the second biotransformation experiment in which single enantiomers were subjected to microbial transformation. These studies were designed to compare the transformation rate of individual iodolactone enantiomers (**1a** and **1b**) as well as the enantioselectivity of the process. The results are presented in Figure 2.

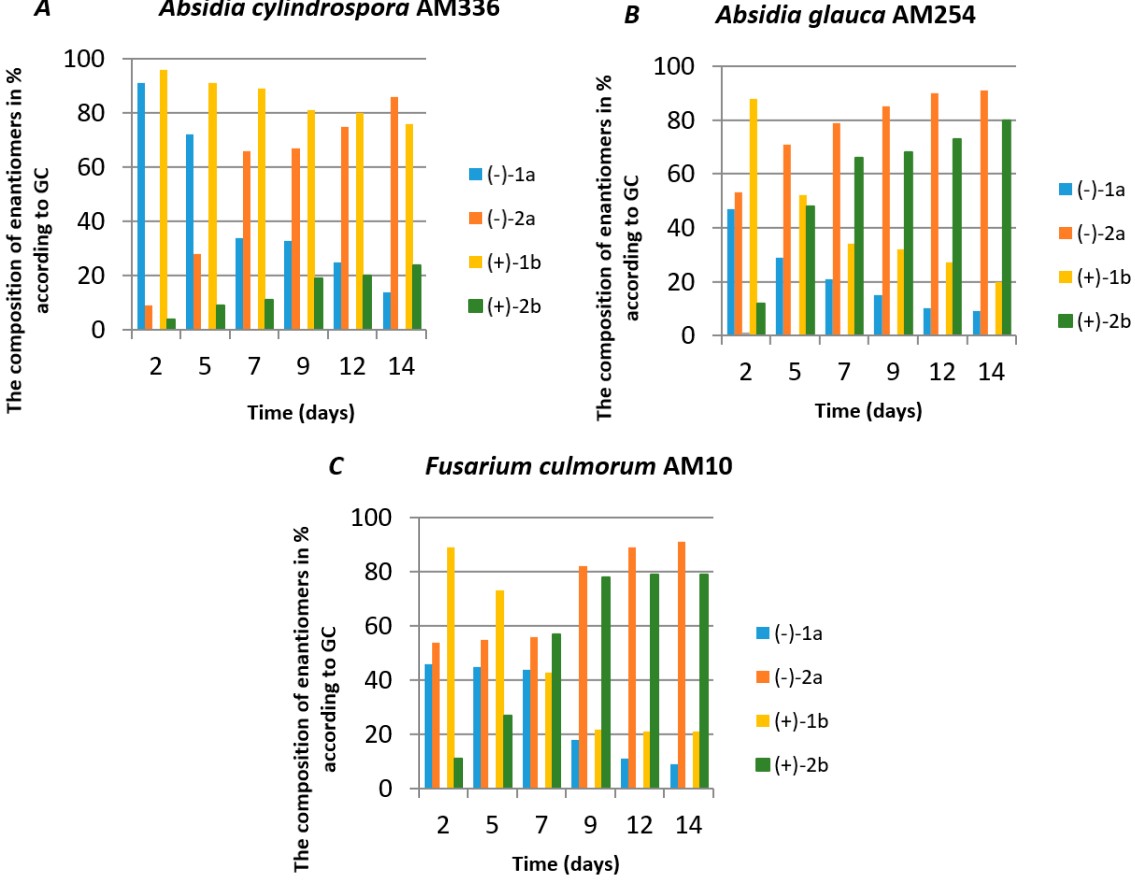

**Figure 2.** Composition (in % according to GC) of the product mixtures in the biotransformations of *cis*-5-(1-hydroxyethyl)-4-phenyldihydrofuran-2-one enantiomers (**1a** and **1b**) by different fungal strains. (**A**) *Absidia cylindrospora* AM336; (**B**) *Absidia glauca* AM254 and (**C**) *Fusarium culmorum* AM10.

Regardless of the biocatalyst used, a single enantiomer of the product was formed from the single enantiomer of the substrate. The enantiomeric excess of hydroxylactones was monitored by chiral GC. The microbial hydrolytic dehalogenation mechanism presented in numerous investigations assumes the inversion of configuration on the halogen-binding carbon atom [21,27,28]. The reason for this is the steric

conditions that determine the approaching nucleophile from the opposite side to the halogen-binding carbon. For reported iodolactones, the formation of hydroxyderivatives with the opposite configuration to the C-6 carbon atom is rational. In these processes, from (−)-(4*R*,5*R*,6*S*)-iodolactone **1a** and (+)-(4*S*,5*S*,6*R*)-iodolactone **1b,** (−)-(4*R*,5*R*,6*R*)-hydroxylactone **2a** and (+)-(4*S*,5*S*,6*S*)-hydroxylactone **2b** were obtained, respectively (Figure 3). Previous studies carried out on cyclic iodolactones also confirm this thesis [22]. In the provided example, biotransformations were performed on bicyclic iodolactones in *Absidia cylindrospora* culture. The iodine atom was in axial orientation on the cyclohexane moiety of the substrate. As a result of hydrolytic dehalogenation processes, the hydroxyl group has an equatorial orientation in the product structure. Therefore, the stereospecificity of this process is also analogical to the chemical $S_N2$ type of substitution.

**Figure 3.** Hydrolytic dehalogenation of *cis*-5-(1-hydroxyethyl)-4-phenyldihydrofuran-2-one enantiomers and predicted configuration of stereogenic centers of products.

Although the transformation rates of iodolactones **1a** and **1b** were slightly different in all experiments, some tendencies can be pointed out. All three tested strains have the ability to transform both enantiomers. At the beginning of the biotransformation process, the **(-)-1a** enantiomer of iodolactone is transformed more effectively compared to **(+)-1b** enantiomer. It can be clearly seen in *Absidia cylindrospora* AM336 and *Absidia glauca* AM254 cultures. Therefore, during the transformation of racemic iodolactone, the enantiomerically enriched product can be obtained depending on the time of the process, especially when *Absidia cylindrospora* AM336 is used as a biocatalyst.

Since β-aryl-γ-lactones are well known for their cytotoxic potential [17,29], in our study, we also plan to evaluate this activity. Depending on the compound structure, the literature provides examples of different biological activities of iodo- and hydroxylactones. The iodolactones with cyclohexane system can exhibit higher biological potential compared to their hydroxy analogs [23,30,31], whereas some hydroxylactones with aromatic rings have similar cytotoxic potential compared to the iodo-derivatives [32]. Therefore, the examination of the cytotoxic activity of biotransformation products was reasonable. An important aspect of these experiments was the comparison of cytotoxicity of single enantiomers. Those experiments were performed on canine leukemia (CLB70) and two canine lymphoma cell lines (CLBL-1, CNK-89). All tested lines were more sensitive to iodolactones **1a** and **1b** than hydroxylactones **2a** and **2b** (Table 2). Interestingly, the (4*S*,5*S*,6*R*)-enantiomer of iodolactone **1b** is slightly more potent towards tested cells than its antipode. This tendency was also observed previously for other structural analogs [17,29]. This confirms that β-aryl-δ-iodo-γ-lactones with the configuration of stereogenic centers 4*S*, 5*S* and 6*R* generally exhibit higher cytotoxicity towards almost all tested cancer cell lines.

**Table 2.** Antiproliferative activity of iodololactones **1a,b,** hydroxylactones **2a,b** and the control—etoposide—against the selected cancer cell lines expressed as IC$_{50}$.

| IC$_{50}$ Values after 72h (µg/mL) | | | | |
|---|---|---|---|---|
| (-)-(4*R*,5*R*,6*S*)-**1a** | (+)-(4*S*,5*S*,6*R*)-**1b** | (-)-(4*R*,5*R*,6*R*)-**2a** | (+)-(4*S*,5*S*,6*S*)-**2b** | Etoposide |
| **CLB70**　61.47 ± 1.23 | 42.23 ± 4.56 | >100 | >100 | 14.31 ± 2.83 |
| **CLBL-1**　40.87 ± 6.09 | 30.19 ± 7.12 | 80.17 ± 9.15 | 71.69 ± 18.57 | 0.02 ± 0.01 |
| **CNK-89**　88.19 ± 9.00 | 80.32 ± 16.77 | >100 | >100 | not investigated |

## 3. Materials and Methods

### 3.1. Analysis

The progress of biotransformations was checked by thin-layer chromatography (silica gel on aluminum plates, DC-Alufolien Kieselgel 60 F254, Merck, Darmstadt, Germany) and gas chromatography (Agilent Technologies 6890N instrument, Santa Clara, CA, USA). The GC analysis was performed on Agilent DB-5HT capillary column ((50%-phenyl)-methylpolysiloxane 30 m × 0.25 mm × 0.10 µm) and hydrogen as the carrier gas. The ee (%) of biotransformation products were determined on the basis of chiral gas chromatography. The CP Chirasil-Dex CB column (25 m × 0.25 mm × 0.25 µm) was used at the following conditions: injector 200 °C, detector (FID) 250 °C, column temperature: 75 °C (1 min), 75–175 °C (1.6 °C × min$^{-1}$), 175 °C (1 min), 175–200 °C (15 °C × min$^{-1}$), 200 °C (2 min).

The purification of the products was carried out using column chromatography on silica gel (Kieselgel 60, 230–400 mesh, Merck, Darmstadt, Germany).

The NMR spectra were performed on a Brüker Avance II 600 MHz spectrometer (Brüker, Rheinstetten, Germany) in CDCl$_3$ solution. The signals at δH = 7.26, δC = 77.16 were used as references.

Jasco P-2000 digital polarimeter (version with iRM controller, Mary's Court Easton, MD, USA) was used to measure the optical rotation. IR spectra were performed on Mattson IR 300 Thermo Nicolet spectrophotometer.

The gas chromatography–mass spectrometry (GC–MS) analysis was performed on a GCMS-SATURN 2000 instrument (Varian, nowadays Agilent, Santa Clara, CA, USA) using a ZB-1 (crosslinked phenyl-methylsiloxane) capillary column (30 m × 0.25 mm × 0.25 µm). The following temperature program was applied: 70 °C (2 min), 75–250 °C (20 °C × min$^{-1}$), 250 °C (3 min), 250–300 °C (20 °C × min$^{-1}$), 300 °C (2 min) (Figure S5).

### 3.2. Substrates for Biotransformation

The iodolactones (**1a,b**) were obtained from benzaldehyde according to procedure described earlier [15,16]. The spectroscopic data of products are in accordance with data described by Gładkowski [15,16] and are presented below to facilitate the spectra analysis and comparison with a biotransformation product:

*cis*-5-(1-Iodoethyl)-4-phenyldihydrofuran-2-one (*rac*-**1**):

[1]H NMR (300 MHz, CDCl$_3$) δ: 2.01 (d, *J* = 6.6 Hz, 3H, CH$_3$-7), 2.71 (dd, *J* = 17.7, 0.9 Hz, 1H, one of CH$_2$-3), 3.14 (dd, *J* = 17.7, 8.7 Hz, 1H, one of CH$_2$-3), 3.47 (dq, *J* = 10.8, 6.6 Hz, 1H, H-6), 3.91 (dd, *J* = 8.4, 5.1 Hz, 1H, H-4), 4.82 (dd, *J* = 10.8, 5.1 Hz, 1H, H-5), 7.25–7.35 (2 m, 5H, -C$_6$H$_5$). [13]C NMR (75 MHz, CDCl$_3$) δ: 23.3 (C-6), 25.5 (C-7), 38.9 (C-3), 45.0 (C-4), 87.8 (C-5), 128.0 (C-4′), 128.6 (C-2′ and C-6′), 128.7 (C-3′ and C-5′), 137.3 (C-1′), 176.5 (C-2). IR (KBr, cm$^{-1}$): 1779, 1416, 1183, 1136, 1005, 755, 706.

### 3.3. Microbial Transformations—Screening Procedure

The biocatalysts tested in the screening procedure were filamentous fungi. The microorganisms derives from the collection of the Institute of Biology and Botany, Wrocław Medical University (AM) (*Trametes versicolor* AM536, *Nigrospora oryzae* AM8, *Mortierella vinaceae* AM149, *Mortierella isabellina* AM212, *Fusarium culmorum* AM10, *Fusarium avenaceum* AM12, *Armillaria mellea* AM296, *Absidia cylindrospora* AM336, *Absidia glauca* AM254). The cultivation of the strains were performed at 20 °C in 300 mL Erlenmeyer flasks containing 50 mL of medium (3% glucose, 0.5% peptone K, 0.5% aminobac in distilled water). The substrate (**rac-1**) was dissolved in 1 mL of acetone and after 3 days of growth added to the shaken cultures (170 rpm). The biotransformations were carried out for 14 days. After 2, 5, 7, 9, 12 and 14 days, the products were extracted with methylene chloride and analyzed by TLC and GC.

### 3.4. Isolation of Obtained Products

Three strains (*Absidia cylindrospora* AM336, *Absidia glauca* AM254, *Fusarium culmorum* AM10) were selected as efficient biocatalyst in transformation processes. For the identification of products and their isolation, the additional experiments were performed. The *Absidia glauca* AM254 strain was grown in six 500 mL Erlenmeyer flasks containing 100 mL of medium. On the third day 20 mg of racemic iodolactone (**rac-1**) were added to each flask. The substrate was dissolved in 1 mL of acetone (total amount 120 mg, conditions the same as described for screening procedure). The product was extracted three times with chloroform (50 mL for each flask). The organic layer was dried over anhydrous magnesium sulfate and evaporated in vacuo. The transformation product was purified by column chromatography (hexane: acetone in gradient from 15:1 to 5:1). As a result the hydroxyderivative (**rac-2**) was obtained.

*cis*-5-(1-Hydroxyethyl)-4-phenyldihydrofuran-2-one (**rac-2**):

Oily liquid, $^1$H NMR: (600 MHz, CDCl$_3$) δ: 1.13 (d, *J* = 6.6 Hz, 3H, CH$_3$-7), 2.85 (dd, *J* = 17.2, 8.9 Hz, 1H, one of CH$_2$-3), 3.06 (dd, *J* = 17.2, 8.3 Hz, 1H, one of CH$_2$-3), 3.58 (qd, *J* = 6.6, 4.3 Hz, 1H, H-6), 3.83 (q, *J* = 8.3 Hz, 1H, H-4), 4.49 (dd, *J* = 7.4, 4.3 Hz, 1H, H-5), 7.27–7.40 (3m, 5H, -C$_6$H$_5$). $^{13}$C NMR (151 MHz, CDCl$_3$) δ: 19.2 (C-7), 35.3 (C-3), 43.9 (C-4), 66.8 (C-6), 86.7 (C-5), 127.9 (C-4′), 128.1 (C-2′ and C-6′), 129.1 (C-3′ and C-5′), 136.9 (C-1′), 176.8 (C-2, C=O). IR (KBr, cm$^{-1}$): 3422, 1773, 1138, 1077, 703.

The biotransformation of (−)-*cis*-(4*R*,5*R*,6*S*)-5-(1-iodoethyl)-4-phenyldihydrofuran-2-one (**1a**) was carried out according to the procedure described above; as a result, the (−)-enantiomer of hydroxylactone was obtained:

(−)-*cis*-(4*R*,5*R*,6*R*)-5-(1-Hydroxyethyl)-4-phenyldihydrofuran-2-one (**2a**)

$[\alpha]_D^{20} = -123.4$ (*c* = 0.75, CHCl$_3$), ee > 99.9%, spectroscopic data identical with those presented for *rac*-**2**.

The biotransformation of (+)-*cis*-(4*S*,5*S*,6*R*)-5-(1-iodoethyl)-4-phenyldihydrofuran-2-one (**1b**) was carried out according to the procedure described above; as a result, the (+)-enantiomer of hydroxylactone was obtained:

(+)-*cis*-(4*S*,5*S*,6*S*)-5-(1-Hydroxyethyl)-4-phenyldihydrofuran-2-one (**2b**)

$[\alpha]_D^{20} = +124.9$ (*c* = 1.4, CHCl$_3$), ee > 99.9%, spectroscopic data identical with those presented for *rac*-**2**.

### 3.5. Antiproliferative Activity

The antiproliferative tests were performed on three types of cancer cell lines: canine B-cell lymphoma (CLBL-1) [33], canine B-cell chronic leukemia (CLB70) [34] and NK-cell lymphoma (CNK-89) (Grudzień et al. in press). The antiproliferative activity was investigated by MTT test after 72 h of treatment, according to the procedure described previously [16]. The concentration of tested lactones was in the range 6.25–50 μg/mL and dimethyl sulfoxide concentration was <1% in each dilution. The optical density of formazan formed in untreated control cells was determined as 100%, and IC$_{50}$

values (μM concentration of the compounds ables to inhibits the proliferation of 50% of cells) were calculated. The results are presented as the means ± standard deviation (SD) of four separate analyses, with three wells each.

## 4. Conclusions

Among ten tested strains of filamentous fungi, *Absidia cylindrospora* AM336, *Absidia glauca* AM254 and *Fusarium culmorum* AM10 have ability to transform *cis*-5-(1-iodoethyl)-4-phenyldihydrofuran-2-one **1**. In all processes, *cis*-5-(1-hydroxyethyl)-4-phenyldihydrofuran-2-one **2** was obtained as an only product. The experiments with the use of single enantiomers, allow as to analyze the enantioselectivity of the process and also predict the configuration of stereogenic centers of the products. The hydrolytic dehalogenation reaction is more likely to occur analogically with chemical $S_N2$ type of substitution, which leads to the inversion of configuration on the halogen-binding carbon atom. For that reason, from (−)-(4*R*,5*R*,6*S*)-iodolactone **1a** the (−)-(4*R*,5*R*,6*R*)-hydroxylactone **2a** was formed and from (+)-(4*S*,5*S*,6*R*)-iodolactone **1b** the (+)-(4*S*,5*S*,6*S*)-hydroxylactone **2a** was formed. The ***rac*-1** was transform with the highest enantioselectivity in *Absidia cylindrospora* AM336 culture. The optimization of this biotransformation may lead to the complete kinetic resolution of iodolactone enantiomers. Under optimal conditions, one enantiomer of the substrate (**1b**) and the enantiomer of hydroxylactone with the opposite configuration of the asymmetric centers on C-4 and C-5 carbon atoms (**2a**) could be obtained. Additionally, the extended screening of microorganisms could result in finding a biocatalyst with higher enantioselectivity or even the opposite enantioselectivity. The evaluation of antiproliferative activity of obtained lactones brings to the conclusion that the activity of the individual enantiomers is not much different. Although, analyzing data presented previously we noticed the tendency that (4*S*,5*S*,6*R*)-enantiomers of β-aryl-δ-halo-γ-lactones are in most cases more potent in cytotoxicity tests. Contrastingly, there is a significant difference between the cytotoxic activity of iodo- and hydroxylactones. The substrates of biotransformation processes were more potent towards all tested cancer cell lines.

**Supplementary Materials:** Supplementary materials can be found at http://www.mdpi.com/2073-4344/10/11/1313/s1. Figure S1: IR spectrum of lactone 2, Figure S2: $^{13}$C-NMR spectrum of lactone 2, Figure S3: $^{1}$H-NMR spectrum of lactone 2, Figure S4: HMBC spectrum of lactone 2, Figure S5: The comparison of GC–MS analysis of iodolactone **1** and hydroxylactone **2**.

**Author Contributions:** Conceptualization, M.M.; Methodology M.M. and A.P.; Investigation, M.M., B.H.-S., A.P., A.K.; Project Administration, M.M.; Validation, B.O.-M. and W.G.; Writing-original draft, M.M.; Writing—Review and Editing, M.M., A.P. and W.G. All authors have read and agreed to the published version of the manuscript.

**Funding:** This research was funded by the statutory activities of the Department of Chemistry, Wrocław University of Environmental and Life Sciences. The Article Processing Charge (APC) was financed under the Leading Research Groups support project from the subsidy increased for the period 2020–2025 in the amount of 2% of the subsidy referred to in Art. 387 (3) of the Law of 20 July 2018 on Higher Education and Science, obtained in 2019.

**Acknowledgments:** We would like to thank B.C. Ruetgen (Institute of Immunology, Department of Pathobiology, University of Veterinary Medicine Vienna) for providing the CLBL-1 cell line. The authors would like to thank Jarosław Popłoński for his support and assistance with HRMS analysis.

**Conflicts of Interest:** The authors declare no conflict of interest.

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
