# Peer review of "Biotechnological Approach for the Production of Enantiomeric Hydroxylactones Derived from Benzaldehyde and Evaluation of Their Cytotoxic Activity"

_catalysts, doi:10.3390/catal10111313_

Round 1

Reviewer 1 Report

The authors in their research article present the analysis of new β-aryl-δ-hydroxy-γ-lactones obtained through the process of biotransformation. Three biocatalysts were shown to be capable of transforming iodolactones, namely  Absidia cylindrospora, Absidia glauca and Fusarium culmorum. The obtained hydroxylactones were evaluated for their cytotoxic activity and displayed lower cytotoxic potential than their substrates. Overall, the manuscript is clearly written and experiments are clearly presented. I have the following comments for the authors regarding their research:

  • Although the approach of the authors relating to the application of biotransformation for the biosynthesis of novel β-aryl-δ-hydroxy-γ-lactone derivatives is interesting, the objectives of the study, as mentioned in the Introduction, were not met. The authors mention the synthesis of new hydroxylactones as potential cytotoxic agents. As shown in the presented results, the obtained hydroxylactones showed lower cytotoxic potential than their substrates. Furthermore, the authors point out that numerous studies indicate the higher biological activity of iodolactones in comparison to their hydroxy analogs. Therefore, what was the objective of the synthesis of these analogs. This requires further clarification.
  • Did the authors mean >100, as in greater than, in Table 1.
  • Revision of the manuscript is required for the correction of some language errors. Examples are listed below:

Line 17: in the biotransformation process new

Line 31: are a group

Line 39: important aspects

Line 44: interesting alternatives

Line 45: the context

Line 47: a great tool

 Line 52: fungal catalysts

Line 52: facilitate

Line 53: sourcing new

Line 57: the preliminary investigation

Line 58: obtain new

Line 65: we planned to further investigate the metabolism of

Line 66: determine the biological potential

Author Response

We are very grateful for Reviewers comments. We hope that our explanations and manuscript corrections will be sufficient and significantly enhance the overall quality of the study.

Reviewer 1

The authors in their research article present the analysis of new β-aryl-δ-hydroxy-γ-lactones obtained through the process of biotransformation. Three biocatalysts were shown to be capable of transforming iodolactones, namely  Absidia cylindrosporaAbsidia glauca and Fusarium culmorum. The obtained hydroxylactones were evaluated for their cytotoxic activity and displayed lower cytotoxic potential than their substrates. Overall, the manuscript is clearly written and experiments are clearly presented. I have the following comments for the authors regarding their research:

  • Although the approach of the authors relating to the application of biotransformation for the biosynthesis of novel β-aryl-δ-hydroxy-γ-lactone derivatives is interesting, the objectives of the study, as mentioned in the Introduction, were not met. The authors mention the synthesis of new hydroxylactones as potential cytotoxic agents. As shown in the presented results, the obtained hydroxylactones showed lower cytotoxic potential than their substrates. Furthermore, the authors point out that numerous studies indicate the higher biological activity of iodolactones in comparison to their hydroxy analogs. Therefore, what was the objective of the synthesis of these analogs. This requires further clarification.

Response: The research presented in the article are part of the structure activity relationship determination for β-aryl-γ-lactones with different substituents in the lactone ring. Whereas, the examples given in the text relate to lactones with cyclohexane system. As the products of biotransformation, to the best of our knowledge are new, the determination of biological potential is crucial. For different systems the activity may relate to various elements of the structure. For that reason, we decided to perform the antiproliferative test. The results support the conclusion that also for this system (β-aryl-γ-lactones) the hydroxy-derivatives are less potent, however they also indicate a difference in activity between enantiomers. In the discussion we pointed out that previous findings on the activity of hydroxylactones relate to structurally different compounds (lines 162-169).

  • Did the authors mean >100, as in greater than, in Table 1.

Response: The table numbering has been improved. The<100 has been corrected to >100.

  • Revision of the manuscript is required for the correction of some language errors. Examples are listed below:

Line 17: in the biotransformation process new

Line 31: are a group

Line 39: important aspects

Line 44: interesting alternatives

Line 45: the context

Line 47: a great tool

 Line 52: fungal catalysts

Line 52: facilitate

Line 53: sourcing new

Line 57: the preliminary investigation

Line 58: obtain new

Line 65: we planned to further investigate the metabolism of

Line 66: determine the biological potential

Response: The suggested changes were made and the manuscript was additionally checked for language correctness.

Reviewer 2 Report

The work is interesting, following on from the authors' extensive previous work in the field.  Although the hydrolytic dehalogenation by these microbial species is not especially new and the inversion of configuration in the transformation  is perhaps not unexpected, the potential for selective transformation of one isomer of the racemic iodolactone is of particular interest.

I would recommend publication after minor revision.

Overall, the manuscript requires improvement in the grammar; there are numerous missing pronouns, and incorrect use of tenses and plurals. These errors hinder the clarity of the argument in some places.

In Table 2, there are errors in the "greater than" signs for compounds 2a and 2b, with IC50 above 100 μg/mL, the "less than", < symbol has been used instead and needs changing to the correct sign, >100.

The heteronuclear correlation (HMBC) is shown in a table - I would expect to also see the original spectral data in the Supporting Information.

The discussion could perhaps benefit from some further comment on whether it is likely that the observed selectivity for transformation of 1a over 1b can be sufficiently optimized with these fungal species or whether a wider species screen is required.

Author Response

We are very grateful for Reviewers comments. We hope that our explanations and manuscript corrections will be sufficient and significantly enhance the overall quality of the study.

Reviewer 2

The work is interesting, following on from the authors' extensive previous work in the field.  Although the hydrolytic dehalogenation by these microbial species is not especially new and the inversion of configuration in the transformation  is perhaps not unexpected, the potential for selective transformation of one isomer of the racemic iodolactone is of particular interest.

I would recommend publication after minor revision.

Overall, the manuscript requires improvement in the grammar; there are numerous missing pronouns, and incorrect use of tenses and plurals. These errors hinder the clarity of the argument in some places.

Response: The language correction has been made.

In Table 2, there are errors in the "greater than" signs for compounds 2a and 2b, with IC50 above 100 μg/mL, the "less than", < symbol has been used instead and needs changing to the correct sign, >100.

Response: Table numbering has been improved. The "greater than"100 has been corrected to "less than".

The heteronuclear correlation (HMBC) is shown in a table - I would expect to also see the original spectral data in the Supporting Information.

Response: The HMBC spectrum was added to the supporting information (Figure S4).

The discussion could perhaps benefit from some further comment on whether it is likely that the observed selectivity for transformation of 1a over 1b can be sufficiently optimized with these fungal species or whether a wider species screen is required.

 Response: The discussion was extended according to the reviewer suggestion (lines 278-284).

Round 2

Reviewer 1 Report

The authors have addressed all of my comments, have made the suggested modifications and provided satisfactory explanations to my questions.

This manuscript is a resubmission of an earlier submission. The following is a list of the peer review reports and author responses from that submission.